# Telemedicine during the Coronavirus Disease (COVID-19) Pandemic: A Multiple Sclerosis (MS) Outpatients Service Perspective



Francesco Corea [1,*], Silvia Ciotti [2], Antonella Cometa [3], Claudia De Carlo [4], Giancarlo Martini [5], Silvano Baratta [2] and Mauro Zampolini [1]

[1]  Neurology Unit, Department of Rehabilitation, Ospedale San Giovanni, Battista di Foligno, 06034 Foligno, Italy; mauro.zampolini@uslumbria2.it
[2]  Department of Rehabilitation, Struttura Complessa Riabilitazione Neuromotoria, 06039 Trevi, Italy; silvia.ciotti@uslumbria2.it (S.C.); silvano.baratta@uslumbria2.it (S.B.)
[3]  Department of Rehabilitation, Santa Maria Stella, 05018 Orvieto, Italy; antonella.cometa@uslumbria2.it
[4]  Department of Rehabilitation, Domus Gratiae, 05100 Terni, Italy; claudia.decarlo@uslumbria2.it
[5]  Department of Rehabilitation, Ospedale di Cascia, 06043 Cascia, Italy; giancarlo.martini@uslumbria2.it
*   Correspondence: francesco.corea@uslumbria2.it; Tel.: +39-074-2339-7063

**Abstract:** Background: During the COVID-19 pandemic, the need for a broader implementation of telemedicine for many diseases has become apparent. Televisits are one type of telemedicine in which clinical visits are conducted remotely using an audio-visual connection with the patient at home. The use of televisits is more established in Stroke care but was also recently formally evaluated for Multiple Sclerosis (MS). This retrospective case series describes patient characteristics and reasons for televisits in persons with MS during the COVID-19 pandemic outbreak in Italy, which was declared in February 2020. Methods: Recruitment occurred in a general hospital based MS clinic during Italy's lockdown months period (9 March–18 May). Each subject completed at least one televisit. The baseline data included were demographics and MS history; reasons for the remote house calls were analyzed focusing on COVID-19 related needs. Results: Forty-six participants completed at least one study visit. The patients enrolled were more often females suffering from Relapsing Remitting Multiple Sclerosis (RRMS). Half of the patients had an intermediate level of education and lived within a 60 min drive from the clinic. These patients predominately had a short disease duration and were mostly involved in oral treatment. The main reasons for the call were drug use and counseling on social distancing. In 5 cases, COVID-19 infection was reported. Conclusions: Televisits during the COVID-19 outbreak demonstrated their utility as a care delivery method for MS. Hence, it is vital to facilitate the implementation of this technology in common practice to both face infectious threats and increase accessibility of the health care system.

**Keywords:** COVID-19; multiple sclerosis; telemedicine

## 1. Introduction

The COVID-19 pandemic reduced the accessibility of many Italian hospitals due to a shortage of resources and a lack of physicians. The disruption of many hospital services was reported with a subsequent need to put adequate countermeasures into place [1–3].

Public health safety measures such as social distancing and postponing non urgent visits triggered a potential decline in the quality and safety of health care. The pharmaceutical distribution for all chronic diseases during the lockdown (e.g., neurological, rheumatological, diabetes) was guaranteed ex-lege. This was planned to minimize the need for access of non immunocompetent patients to health care facilities during the lockdown. As a result, even if a medical prescription was overdue, patients were allowed to continue long-term therapies. Drug delivery was also provided to MS patients under all Disease Modifying Drugs (DMTs). This was needed since the access to neurology services was also restricted

to urgent cases only during the lockdown. Such an emergency procedure raised safety concerns in the medical community towards potential harmful medications misuses.

Telemedicine solutions were found to be feasible and cost-effective in many neurological diseases, as well as Multiple Sclerosis (MS) [4–7]. Specific, quickly deployable, and mobile telemedicine solutions may increase the access to care for patients with mobility limitations or geographic barriers [8].

The aim of this retrospective case series analysis was to evaluate the use of mobile telemedicine in a MS outpatients service during the COVID-19 pandemic lockdown. The population demographic data are described together with disease characteristics, type of treatment, and main reason for the televisits.

## 2. Methods

All of the patients of our outpatients service (total population of 300 subjects assisted) needing a follow up visits during the lockdown months period (March 9th–May 18th) were candidates.

Participants were recruited from the local MS database. Inclusion criteria included age >18 and a diagnosis of MS (relapsing or progressive). Participants were initially required to have an internet connected smartphone or tablet. Participants were required to reside within the same region. Exclusion criteria included a lack of proficiency in written or spoken Italian or a lack of coordination ability (or caregiver assistance) to manipulate the internet-connected device.

The patients, after a screening selection, received a telephone call in which a telemedicine visit was offered instead of the standard in person visit.

*Technical Aspects*

The software adopted for the purpose was a mobile health app: Meydoc ® by Meytec Gmbh Werneuchen (Berlin, Germany). It was selected based on ease of use and compliance with legal European Community and National policies.

For the telemedicine visits, participants received e-mail links prior to the visit with an activation PIN and a link to automatically download the app and allowed them to connect to their specialist at the scheduled time. The e-mail also provided participants with a phone number to contact their MS neurologist directly if there were difficulties establishing a connection. The neurologist had the ability to obtain support from information technology staff 24/7 based on when needed on a call. For the telemedicine visits, participants completed a "test connection" a short check of the video/audio connection with the specialist a few days prior to the subject's telemedicine visit (Figure 1 describes the organization of the system).

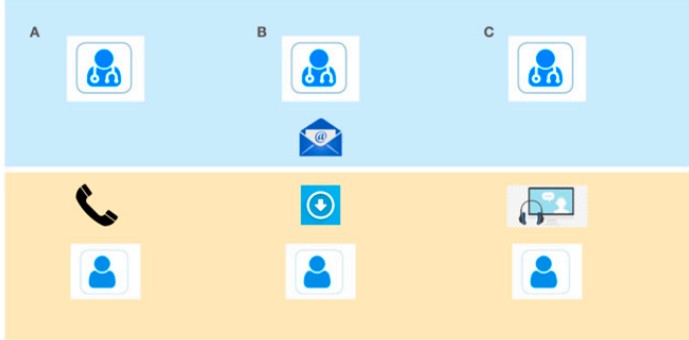

**Figure 1.** Enrollment scheme. (**A**) telephone contact and inclusion of the subject in the program. (**B**) e-mail with link to download the app and activation PIN. (**C**) Televisit and/or technical test.

At each visit, participants provided interval clinical history and underwent a focused general neurological examination derived from the Italian multiple sclerosis register framework. The register methodology is reported elsewhere, but the neurological evaluation

is focused on the rating of the Expanded Disability Status Scale (EDSS) using a standard operating procedure. (Troiano 2019). After each visit, participants were emailed reports that were specific to the visit type.

The physician in the MS clinic used a standard PC on which the master license of the software was downloaded. The overall setting of the system took less than 24 h and needed minor technical support.

## 3. Results

### 3.1. Subject Participation

Participants were recruited over 8 weeks, and a total of 46 participants joined the program (15.3% of the outpatients population, 76.6% of monthly usual visit volume). At least 1 study visit was completed by all participants and 30 participants completed multiple visits. In 2 cases, the quality of the connection was low because of low signal (using mobile telephone lines). Figure 2 shows a general outlook of the patients enrolled; many of the patients were in the same county to the hospital, while others were from other counties of central Italy, 1 was from northern Italy, and 1 was from central Asia.

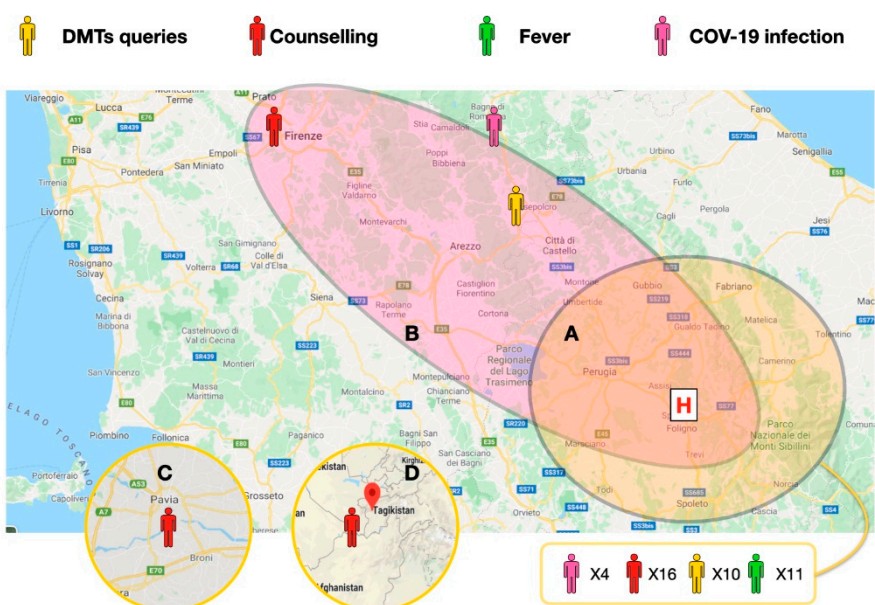

**Figure 2.** Central Italy map: Umbria, Marche, Toscana, Emilia Romagna. Patient localization and driving distances from the MS clinic. (**A**) Within a 60 min drive. (**B**) within a 120 min drive. (**C**) and (**D**) Cases in Lombardy and central Asia. The main reasons for the televisits in the 46 cases. The defined COVID-19 cases localization is depicted in pink; fever of unknown causes is depicted in green; Disease Modifying Drugs (DMTs) related queries and counseling for social distancing are depicted in yellow and red, respectively. All the calls in the closer areas are reported in a single box (41 cases).

### 3.2. Demographics

Table 1 summarizes the characteristics of the 46 study participants who completed at least one televisit. The mean subject age was 42.8 with a female predominance (65.2%). A large majority of participants (86.9%) had a Relapsing Remitting form of MS with a shorter disease history (below 10 years: 86.9%). The mean duration time of the disease in the population was 8.9 years.

**Table 1.** Participant characteristics.

| | |
|---|---|
| **Age, Mean Years (SD)** | 42.8 (10.8) |
| Gender, female | 30 (65.2%) |
| **Race** | |
| Caucasian | 46 (100%) |
| Other | 0 |
| **MS Type** | |
| Relapsing remitting | 40 (86.9%) |
| Secondary progressive | 6 (13.1%) |
| **Disease Duration** | |
| <10 years | 40 (86.9%) |
| ≥10 and ≤20 | 5 (10.8%) |
| >20 years | 1(2.1%) |
| **Disease Modifying Therapy** | |
| Injectable | 16 (34.7%) |
| Oral | 20 (43.4%) |
| IV | 10 (21.7%) |
| None | 0 |
| **Education** | |
| High school graduate | 8 (17.3%) |
| College graduate | 30 (65.2%) |
| Higher degree | 8 (17.3%) |
| **Income** | |
| <25,000 | 0 |
| 25,000–49,999 | 31 (67.3%) |
| 50,000–99,999 | 15 (32.6%) |
| >100,000 | 0 |
| Declined | 0 |
| Employed | 30 (65.2%) |
| **Drive Distance Time** | |
| 0–30 min | 20 (43.4%) |
| 31–60 min | 20 (43.4%) |
| 61–120 min | 3 (6.5%) |
| >120 min | 3 (6.5%) |

An intermediate level of education was reported in 65.2% of subjects. The annual income reported in 67.3% of cases was between 25,000 € and 50,000 €. A total of 43.4% of cases were treated with oral DMTs, while injectables were used in 24.7% of cases and i.v. drugs were used in 21.7% of cases.

A total of 87% of the participants lived less than a 60-min drive from the hospital, with 13% living more than 2 h away.

### 3.3. Reasons for Televisit

The main reasons for the telemedicine visit were (Table 2) DMTs use during the pandemic and counseling for social distancing. In all cases, these topics were discussed

with the patients themselves and often with their families. Other frequent needs were: advice for fevers of an unknown cause (21.7%), routine magnetic resonance imaging (MRI) re-scheduling (54.3%). A total of 5 subjects had COVID-19 infection diagnosed (10.8%) later on with documented molecular testing via the nasofaringeal swab. Other reasons (3 cases: 6.5%) were counseling for pregnancy, driving license advice, and anti epileptic drug titration.

**Table 2.** Main reason/complaints of patients seen in televisits.

| Main Complaint | DMTs Use /Dosage | Fever of Unknown Cause | COVID-19 Infection | Counselling on Social Distancing | Imaging | Others |
|---|---|---|---|---|---|---|
| Number of patients | 46 | 10 | 5 | 46 | 25 | 3 |
| % | 100 | 21.7 | 10.8 | 100 | 54.3 | 6.5 |

Patients who participated in multiple calls (30 cases) were: those complaining of fevers of an unknown cause (10 cases), patients suffering COVID-19 infection (5 cases), and those needing drug dose changes involving oral DMTs (15 cases). No patients completed more than 2 televisit calls.

## 4. Discussion

The COVID-19 pandemic required rapid adoption of new technologies to improve access to healthcare while social distancing was mandatory. The current COVID-19 pandemic with many associated local and national restrictions have compelled healthcare systems across the world to adopt telemedicine at unprecedented speed.

Our MS service implemented telemedicine as a way to continue to provide care to our patients during the crisis. All patients contacted for telemedicine visits showed interest in and a willingness to use this modality. Televisits were found to be feasible and engaging to both persons with MS and physicians.

Due to the technology requirements, our concern is that some telemedicine solutions may potentially exacerbate pre-existing disparities for access to high profile care. As far as we observed, the larger availability of low cost but efficient smartphones seems to reduce this potential gap. Both low-educated and unemployed groups are consistently represented in our survey. A family income of between 25,000 € and 50,000 € may even be considered very low depending on the size of the family. This could not be addressed in our study.

We reported, as expected based on the prevalence of MS, a female gender predominance as well as a larger presence of Relapsing Remitting Multiple Sclerosis (RRMS) forms. Concerning data quality of the case series, we can report how our clinic is collaborating with the national MS database network in providing a single standardized protocol under permanent quality monitoring [9].

To the best of our knowledge, even if on a modest sample size, this is the first real life case series in continental Europe assessing mobile phone televisits involving MS. Similar experiences of telemedicine adopted unspecified technologies, [10] or included smaller samples of selected patients. Others were dedicated to rehabilitation of selected MS patients [11–13].

Another notable element is related to the type of geographical area taken into consideration. Our practice is situated in a fairly large (8546 km$^2$) and mainly rural area with a county population of half a million (mean density: 103 people per km$^2$). Technology usage is expected to be low in terms of internet access and broadband connections in this area.

We reported a low quality of connection in 2 cases due to unstable mobile signal quality. This was reported in televisits for those living in isolated mountainous areas (internal areas of the Orvieto municipality and Appennini mountain areas). From the physician perspective, these televisits were considered to be informative to determine whether follow up visits were needed. We unfortunately had not implemented any systematic analysis

of the patients' perception at that time. This happened because the televisit program was undertaken as an urgent countermeasure to the pandemic emergency. Patients should have filled a satisfaction questionnaire on the service and quality of the video call. We may take into account that fact that we used an end-to-end and serverless system. As a result of this, we would have asked patients to give an opinion on their own domestic internet contract and on the quality of their smartphone camera. The overall response of the community to the program was positive, and will be extended as a permanent service.

We evaluated using the televisit program for a large majority of the monthly scheduled outpatients. Televisits during the analysis period were offered to those with overdue prescriptions. Another group of patients undergoing visits for other reasons were thereby postponed. No subject refused to join the project when it was proposed.

Although some neurological exam modifications or substitutions were utilized, the televisits provided similar information compared with standard visits. For some physicians, there was an extra value given by the observation of a patient's home environment [14]. The ease of connectivity was appealing for physicians as well as for participants. Other physicians determined that higher camera resolution aided in certain exam components such as tracking eye movements and observing rapid movements. We have no data concerning the resolution of the smartphones. The use of common tools for neurological examination demonstrated a possible ability to improve tele-neurology practices [15].

Also, EDSS was previously demonstrated as being feasible using telemedicine, with more consistent assessments used for optic, bowel and bladder, and cerebral functions. The least consistent assessments were for cerebellar and brain stem functions. Agreement between the remote and local examiners was demonstrated as being similar to that reported for different neurological examiners directly assessing the same patient [16]. The further validation of neurological impairment and outcome measures was beyond the scope of our experience.

The intervention of the neurologist was also relevant in cases of fevers of an unknown cause and in the dispatchment of suspect COVID-19 cases. Four of the patients were on an oral DMTs regime, while one was involved in injectable interferon therapy. Hospitalization was just needed in one case with DMT discontinuation. All of these patients had full recovery.

The COVID-19 challenge forced the minimizing of contact between different people on the one hand and at the same time led to an increasing number of patients in need of specific attention on the other. Since contact reduction is difficult to reduce within the healthcare sector, an unsustainable number of infected professionals would reduce the capacity of MS clinics.

Telemedicine is an affordable solution in major emergencies, as well as in more ordinary scenarios. It should be encouraged in all areas of neurology where remote treatment has been shown to be feasible and effective [17].

**Author Contributions:** F.C. was the main contributor to the writing of this manuscript. S.C., A.C., C.D.C., S.B., G.M., M.Z. contributed in dataset management and patients enrollment. All authors have read and agreed to the published version of the manuscript.

**Funding:** There was no funding received for this manuscript.

**Institutional Review Board Statement:** Comitato Etico Regionale Umbria approved the registry. Reference: 9050/16/ON, Sept 2016.

**Informed Consent Statement:** All patient gave written informed consent to their inclusion in the database.

**Acknowledgments:** The authors would like to thank Zara Erliz for providing editing assistance.

**Conflicts of Interest:** The authors declare no conflict of interest.

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
