# Peer review of "Telemedicine during the Coronavirus Disease (COVID-19) Pandemic: A Multiple Sclerosis (MS) Outpatients Service Perspective"

_2035-8377, doi:10.3390/neurolint13010003_

Round 1

Reviewer 1 Report

The objective of the paper is original and in the context of the group’s and the journal’s focus well placed. The study is focused on the important question of how to implement telemedicine approaches to facilitate care for neurological patients in times of the (COVID-19) pandemic. This is not only in the scope of the journal , but also fits in the context of former publications of the group. The MS cohort is of special interest due to the long-term need for disease modifiying therapy (DMT) in these patients together with the necessity to adopt the DMT according to the course of the disease. Telemedicine could be very useful to overcome pandemia associated restrictions in out-patients (neurology) clinics. This is now shown in this helpful study of Corea et al.

Points to reconsider:

Figure 1: line 83 C: Televisit…..instead of B:...

Figure 2 does not completely fit to its legend nor to the numbers in table 2.

line 85: what exactly does focussed general neurological examination mean. I gather a syndrome oriented audio-video-based examination was performed. Are there reliable data on the use of scoring systems in context of telemedicine and MS as compared to NIHSS and tele-stroke-evaluation?  

Results: the complete number of televisits for the 46 patients would be of interests as well as the reasons for follow-up-visits since 30 patients underwent multiple visits.

Discussion: the physicians perspective on the televisits was discussed, even though not systematically evaluated or stated upon in the results. Furthermore the patients perspective would have been also of interest to the readers.

References: citations need some corrections in the text including spelling errors.

Author Response

Reply to the reviewer #1

1-The legend of Figure 1, was changed with B instead of C. 

2-The Figure 2 was updated for a clearer view. All patients in the closer urban area are in single box.

3- line 85: what exactly does focussed general neurological examination mean. I gather a syndrome oriented audio-video-based examination was performed. Are there reliable data on the use of scoring systems in context of telemedicine and MS as compared to NIHSS and tele-stroke-evaluation?  

Thank you for your comments. The question you raised was very valuable. The neurological examination performed for our patients was based on the Italian MS registry framework, described in a paper reported as reference (troiano 2019). The checklist adopted offers a wide EDSS based assessment. The EDSS was demonstrated being feasible via telemedicine (Kane 2008) it was beyond the scope of this study to further validate neurological impairment and outcome, but I included a new sentence and reference to help the reader on the topic.

4- Results: the complete number of televisits for the 46 patients would be of interests as well as the reasons for follow-up-visits since 30 patients underwent multiple visits.

Thank you for your comments. The question you raised was very valuable. We included a description of the multiple calls.

5-Discussion: the physicians perspective on the televisits was discussed, even though not systematically evaluated or stated upon in the results. Furthermore the patients perspective would have been also of interest to the readers.

Thank you for your comments. This point needs to be better clarified. I included in the discussion the reasons for not having done a patients satisfaction questionnaire. First the urgent need of patients coverage during the lockdown. The limits of similar surveys are also discussed since patients were using their own personal devices and commercial internet traffic contracts. The large diffusion of video calls for disabled people make somehow trivial to ask if they appreciated such approach. People are more ready for telemedicine than are the health service providers. The perception I got from many patients was "finally you made it !". To tell the truth I felt out of date for not having done this years ago.

Reviewer 2 Report

This is a quite nice paper facing a current topic. The authors described their telemedicine experience with MS patients during the coronavirus lockdown, coming to conclusions of a feasible and useful modality.

In my opinion, the manuscript is suitable for publication in Neurology International, after the authors have addressed the following comments and questions:

  • "At each visit, participants provided interval clinical history and underwent a focused general neurological examination" (line 84). Please be more specific. It might be interesting to describe what kind of neurological evaluation was carried out, to report whether new neurological deficits have been documented and if any televisits required subsequent face-to-face visits. 
  • it would be interesting to know the characteristics of those patients who refused to participate in the telemedicine program and for what reasons. please comment on this in the discussion.
  • The paper is generally well written and structured, however the text must be checked for typos.
  • The following sentence is not clear: "All the patients contacted for telemedicine visits showed interest and willingness to develop treatments with this modality" (line 133). Please clarify it.

Author Response

  1. "At each visit, participants provided interval clinical history and underwent a focused general neurological examination" (line 84). Please be more specific. It might be interesting to describe what kind of neurological evaluation was carried out, to report whether new neurological deficits have been documented and if any televisits required subsequent face-to-face visits. 

Thank you for your comments. The question you raised was very valuable. We updated the description of the Televisits reporting an EDSS based analysis. In references we add the Italian MS Registry framework to help readers to focus the standard operating procedure adopted. I also add a reference regarding EDSS validation in telemedicine. 

2. it would be interesting to know the characteristics of those patients who refused to participate in the telemedicine program and for what reasons. please comment on this in the discussion.

Thank you for your comment this point needs to clarified. No patients refused to join the project.We evaluated using the Televisit program a large majority of the monthly scheduled outpatients. Televisits during the analysis period were offered to those with overdue prescriptions. Another group of patients, undergoing visits for other reasons, were thereby postponed.

3. The following sentence is not clear: "All the patients contacted for telemedicine visits showed interest and willingness to develop treatments with this modality" (line 133). Please clarify it.

Thank you the sentence was changed in "All patients contacted for telemedicine visits showed interest and willingness to use this modality."